# Research on a Visual Servoing Control Method Based on Perspective Transformation under Spatial Constraint

**Chenguang Cao** 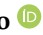

School of Electrical and Electrical Engineering, Shandong University of Technology, Zibo 255000, China; guangcc@foxmail.com

**Abstract:** Visual servoing has been widely employed in robotic control to increase the flexibility and precision of a robotic arm. When the end-effector of the robotic arm needs to be moved to a spatial point without a coordinate, the conventional visual servoing control method has difficulty performing the task. The present work describes space constraint challenges in a visual servoing system by introducing an assembly node and then presents a two-stage visual servoing control approach based on perspective transformation. A virtual image plane is constructed using a calibration-derived homography matrix. The assembly node, as well as other objects, are projected into the plane after that. Second, the controller drives the robotic arm by tracking the projections in the virtual image plane and adjusting the position and attitude of the workpiece accordingly. Three simple image features are combined into a composite image feature, and an active disturbance rejection controller (ADRC) is established to improve the robotic arm's motion sensitivity. Real-time simulations and experiments employing a robotic vision system with an eye-to-hand configuration are used to validate the effectiveness of the presented method. The results show that the robotic arm can move the workpiece to the desired position without using coordinates.

**Keywords:** spatial constraint; visual servoing; perspective transformation

## 1. Introduction

Image-based visual servoing (IBVS) is a humanoid control method for a robotic arm [1]. The main purpose of IBVS is to design a global controller that employs visual feedback signal to generate a screw velocity as the control input for the robotic arm, resulting in the desired joint velocity. As a result, the robotic arm will become faster and more dexterous in applications such as robotic assembly, unmanned aerial vehicles [2], and robotized tracking [3,4]. However, the relationship between the robotic arm's motion and the evolution of visual input is non-linear. In contrast, the model used in a controller is a linearization result of the robotic system, making it difficult to respond effectively to some unique circumstances. For instance, it is not easy to obtain an effective control signal for the robotic arm using existing methods without complete information [5]. Furthermore, another well-known problem emerges in the case of significant rotation about the target, where the target moves away from the desired position and then back again [6]. This phenomenon may cause the target to move out of view (FOV). However, to calculate the feedback signal during the servoing process, the target must be confined inside the FOV of the camera. Using classical control theories, it is not easy to achieve, especially when substantial rotational motion is involved [7–9]. Several constraints must be imposed on the controller to ensure that the robotic arm follows the desired trajectory to avoid the scenarios described above.

The constraints, which refer to the limitations imposed on the motion of the robotic arm by the camera's FOV and equipment requirements, are issues to consider when designing a visual servoing controller. Two widely used techniques for incorporating constraints into controllers are model predictive control (MPC) and trajectory planning [10,11]. The MPC

is primarily used in the field of industry. Its premise is to solve an online finite horizon open-loop-constrained optimization problem by combining acquired image information with system constraints [12]. The resulting sequence is then considered the system control signals. An MPC method based on the discrete-time visual servoing model was published in [13–16] to obtain the convergence of robot motion by non-linear constraint optimization. The MPC method allows the robotic arm to move closer to a straight line, keeping the end-effector in the camera's field of view and preventing the controller from generating signals that violate physical constraints [17]. Moreover, the control signal produced by the MPC controller cannot cause robotic joint limits to be exceeded [18]. However, in the presence of obstacles, the controller based on MPC cannot actively modify the trajectory, implying that the robotic arm cannot be driven to follow a non-linear trajectory.

The other approach for effectively solving the constraint problem is trajectory planning, which involves creating an executable image trajectory within the constraints of a dynamic, unstructured environment, and then driving the robotic arm along the planned trajectory to accomplish the task [19–21]. Trajectory planning methods employed in visual servoing can be classified into three categories based on the features and assumptions [22]. The first one is to generate an image trajectory initially using polar geometry or projective geometry, then a controller drives the robotic arm along this image trajectory [23–25]. The end-effector can travel in a straight line in the workspace and is always in the FOV of the camera, thereby dealing with the issue of system constraint. This method, however, is not always practical, and it is challenging to create a trajectory when we are unable to obtain the projection of the desired position. The second method employs a potential artificial field [26]. The controller uses the sum of potential energy to drive the robotic arm away from the obstruction and toward the target position [27]. It is worth noting that different approaches are required to overcome the problem of local minimization. The final one is to use optimization algorithms to find an optimized spatial trajectory for the robotic arm to reach the target position swiftly [28–30]. On the other hand, the optimization-based approach needs precise environmental data and camera models. In many cases, this is impossible to achieve [31].

It should be noted that when a robotic arm is required to transfer a workpiece to an unmarked position under uncalibrated conditions, these existing approaches are ineffective owing to a lack of critical position information. This work first establishes a geometric model to describe the spatial constraint. Then a visual servo control approach based on perspective transformation is proposed to handle the problem successfully. The central concept of the proposed method is to create a two-dimensional virtual image plane and then project all the targets in the plane. By tracking the image features in the virtual image plane, the workpiece will be moved to the unmarked desired position. The rest of this research is organized as follows. In Section 2, a geometric model of spatial constraint is developed to forecast the future behavior of a robotic arm. Then, a visual servo control method based on perspective transformation is proposed in Sections 3 and 4. The simulations and experiments are conducted in Sections 5 and 6 to demonstrate the effectiveness of the proposed method. Finally, Section 7 draws conclusions.

## 2. Problem Statements

Two automated assembly procedures with a robotic arm are depicted in Figure 1. An air compressor assembly in an outside air conditioning production line is depicted in Figure 1a Four pre-drilled screw holes in the air compressor should be passed vertically through the stud on the base plate to finish the assembly. Figure 1b depicts a rectangular workpiece assembly that must be inserted into a slot.

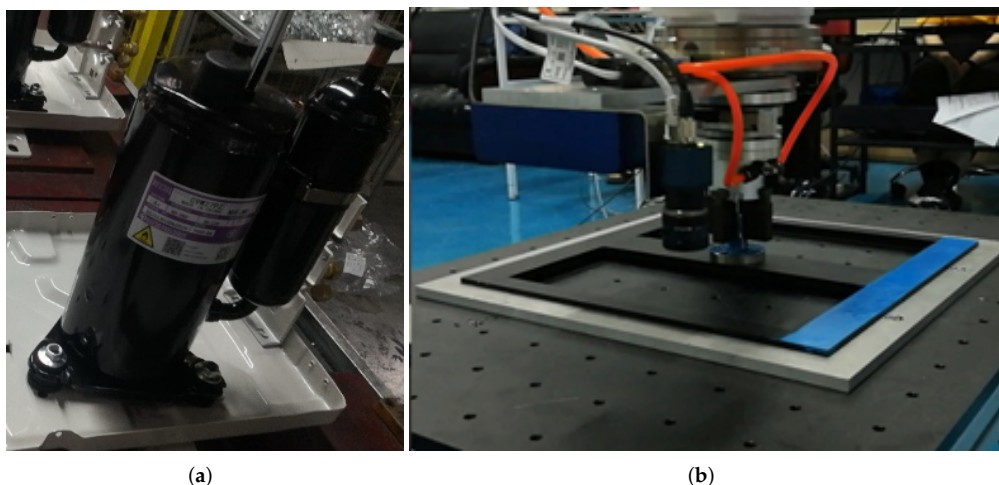

<div align="center">(<b>a</b>)        (<b>b</b>)</div>

**Figure 1.** Two examples of robot-based assembly. (**a**) An air compressor assembly. (**b**) A rectangular workpiece assembly.

A feature that all of the items represented in Figure 1 have in common is that the interface between the workpieces and the base plates has been designed in a unique form to ensure the robustness of the products. As a result, the workpieces should be moved to a spatial node in a confident attitude and then moved perpendicular to the base plates to finish the assembly. The constraint imposed by the interface can be defined as a spatial constraint. The spatial node satisfying the spatial constraint can also be regarded as an assembly node. Without calibration, it is difficult for an IBVS controller to obtain any position information about the assembly node, making it difficult for the robotic arm to reach the assembly node precisely. As seen above, while developing an IBVS assembly controller, the issue of spatial constraint must be addressed.

A more intuitive geometric model was constructed to better illustrate the spatial constraint in IBVS, as shown in Figure 2. There is a purple workpiece with three through holes and a base plate with three columns. The holes have the same spatial relationship as the columns, allowing the workpiece to be securely connected to the base plate. Additionally, $\Re_1$, $\Re_2$, and $\Re_3$ are three assembly nodes, each with its own individual curve. The yellow curve depicts the intended trajectory of the robotic arm. In contrast, the green curve indicates the actual trajectory of the robotic arm. While the yellow curve is the most energy-efficient path between the workpiece and the base plate, it does not meet the spatial constraint. In comparison, the red-blue trajectories, while longer, are consistent with our expectations for this assembly task. The spatial constraint could be repeated. When a workpiece reaches the assembly nodes $\Re_i$ confidently, it must be moved closer to the base plate in a perpendicular direction to complete the assembly task.

Notably, the number of assembly nodes is not constant, resulting in diverse assembly trajectories. These trajectories pass through the assembly nodes $\Re_i$, independent of their morphologies. Each assembly trajectory can be divided into two parts, denoted by the blue and red curves. These parts can be referred to as the transfer and docking trajectories. The shape of the transfer trajectory is unknown, whereas the docking trajectory is a straight line. Suppose the workpiece can be brought smoothly to the assembly node confidently. In that case, the workpiece and base can be docked successfully within the attitude limitation. The image features of assembly nodes, on the other hand, cannot be retrieved without calibration, as assembly nodes lack apparent identity in space.

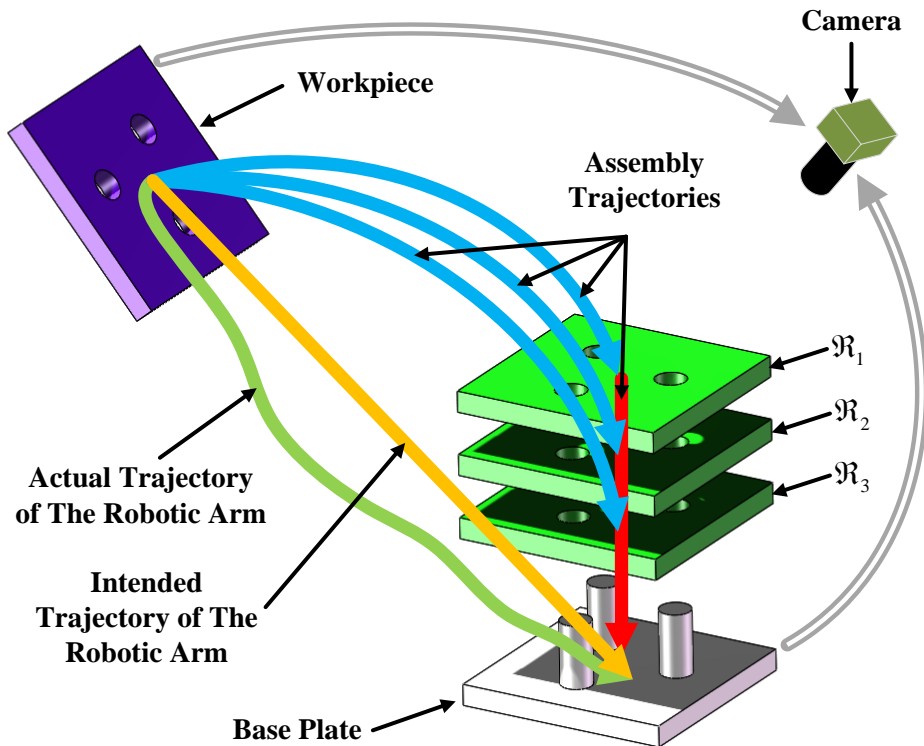

**Figure 2.** Schematic diagram of assembly nodes.

## 3. Visual Servoing Control Method Based on Perspective Transformation

### 3.1. Methodology

A visual servo control method based on perspective transformation is presented in this research to overcome the issue of spatial constraint in IBVS. The method is divided into three steps, which are detailed below.

(1) A virtual image plane $\widetilde{\gamma}$ is generated, and then two homography matrixes $H^{\alpha}$ and $H^{\beta}$ are established.

(2) Assuming that $p_i^{\alpha}$ and $p_i^{\beta}$ are the projections of spatial points $P_i^{\alpha}$ and $P_i^{\beta}$, respectively, in an image. Then, using matrix $H^{\alpha}$ to map $f_i^{\alpha}$ into the virtual image plane $\widetilde{\gamma}$, a new feature $\widetilde{f}_i^{\alpha}$ is created. In the same way, mapping $f_i^{\beta}$ into the virtual image plane $\widetilde{\gamma}$ with $H^{\beta}$ yields a new feature $\widetilde{f}_i^{\beta}$. When $\widetilde{f}_i^{\alpha} = \widetilde{f}_i^{\beta}$, we believe that $P_i^{\alpha}$ deviates from $P_i^{\beta}$ exclusively in the direction of the $Z$-axis. If $F_i^{\alpha}$ represents a set of feature points on the workpiece and $P_i^{\beta}$ represents the corresponding feature points on the base plate, when $\widetilde{f}_i^{\alpha}$ equals $\widetilde{f}_i^{\beta}$, the workpiece has already arrived at the assembly node.

(3) Assuming that the workpiece is located on the end-effector of a robotic arm. After that, the attitude of the end-effector is extracted, and the robotic arm is driven along a linear trajectory under the attitude, thereby docking the workpiece with the base plate.

### 3.2. Feasibility Analysis

The model of perspective transformation is shown in Figure 3. The world frame $R_r$ is composed of the axes $X_r$, $Y_r$, and $Z_r$ and the point $O_r$. $\alpha$ and $\beta$ are two planes which equations are shown in (1) and (2), respectively.

$$Z = Z^{\alpha} \tag{1}$$

$$Z = Z^{\beta} \tag{2}$$

where $Z_\alpha = \Delta Z + Z_\beta$. In Figure 3, there are three cameras with the same internal matrix $M_c$, which defined in (3).

$$M_c = \begin{bmatrix} f/dx & 0 & u_0 \\ 0 & f/dy & v_0 \\ 0 & 0 & 1 \end{bmatrix} \tag{3}$$

where where $f$ is focal length, $dx$ and $dy$ are the distances between adjacent pixels in the $u$ and $v$ axes, respectively. $u_0$ and $v_0$ are row and column numbers of the center.

The frame $R_s$ of camera $s$ is composed of $X_s$, $Y_s$, and $Z_s$ axes and the point $O_s$. $Z_s$ is not parallel to the planes. The frame $R_{e_1}$ of camera $e_1$ is composed of the axes $X_{e_1}$, $Y_{e_1}$, $Z_{e_1}$, as well as the point $O_{e_1}$. The frame $R_{e_2}$ of camera $e_2$ is made up of the axes $X_{e_2}$, $Y_{e_2}$, $Z_{e_2}$ and the point $O_{e_2}$. Both $Z_{e_1}$ and $Z_{e_2}$ are parallel to $\alpha$ and $\beta$. $I_{e_1}$, $I_{e_2}$ and $I_s$ are three image planes. $\tilde{P}_i^r$ denotes a spatial point, which can be expressed by $P_{e_1}$ and $P_{e_2}$ in the frame $R_{e_1}$ and $R_{e_2}$. The relation between $P_{e_1}$ and $P_{e_2}$ can be expressed as (4).

$$P_{e_2} = H_{e_1}^{e_2} P_{e_1} \tag{4}$$

where

$$H_{e_1}^{e_2} = \begin{bmatrix} 1 & 0 & 0 & 0 \\ 0 & 1 & 0 & 0 \\ 0 & 0 & 1 & \Delta Z \\ 0 & 0 & 0 & 1 \end{bmatrix} \tag{5}$$

It is self-evident that for any point $P_i^\beta$ in the plane $\beta$, a corresponding point $p_i^\beta$ in the image plane $I_s$ must exist, and the relationship between $P_i^\beta$ and $p_i^\beta$ can be expressed as (6).

$$p_i^\beta = \xi_i^\beta M_c H_s^r P_i^\beta \tag{6}$$

where $\xi_i^\beta$ is a scale factor. The projection points $p_i^{e_1}$ of $p_i^\beta$ in $I_{e_1}$ are given by (7).

$$p_i^{e_1} = \frac{H_{e_1}^s p_i^\beta}{\xi_i^{e_1}} \tag{7}$$

where $H_{e_1}^s$ is a homography matrix and $\xi_i^{e_1}$ is a scaling factor. Similarly, for point $\tilde{P}_i^r$, a corresponding point $\tilde{p}_i$ in the image plane $I_s$ must exist, and the relationship between $\tilde{P}_i^r$ and $\tilde{p}_i$ can be represented as (8).

$$\tilde{p}_i = \xi_i^\alpha M_c H_s^r \tilde{P}_i^r \tag{8}$$

where $\xi_i^\beta$ is another scaling factor. (9) can be used to describe the projection points $p_i^{e_2}$ of $\tilde{p}_i$ in $I_{e_2}$.

$$p_i^{e_2} = \frac{H_{e_2}^s \tilde{p}_i}{\xi_i^{e_2}} \tag{9}$$

where $H_{e_2}^s$ is a homography matrix. If $p_i^{e_1}$ equals $p_i^{e_2}$, then (10) can be derived by combining (7) and (9) and simplifying.

$$\xi_i^{e_2} \xi_i^\alpha H_{e_2}^s M_c H_s^r P_i = \xi_i^{e_1} \xi_i^\beta H_{e_1}^s M_c H_s^r P_i^\beta \tag{10}$$

Assuming that $H_{e_1}^s$ and $H_{e_2}^s$ are selected as

$$\begin{cases} H_{e_1}^s = (M_c H_{e_1}^r)(M_c H_s^r)^{-1} \dfrac{\xi_i^{e_2}}{\xi_i^\beta} \\ H_{e_2}^s = (M_c H_{e_2}^r)(M_c H_s^r)^{-1} \dfrac{\xi_i^{e_1}}{\xi_i^\alpha} \end{cases} \tag{11}$$

Substituting (11) into (10) yields

$$H^r_{e_2} \tilde{P}^r_i = H^r_{e_1} P^\beta_i \tag{12}$$

where $H^r_{e_1}$ is a transformation matrix between the frames $R_{e_1}$ and $R_r$, $H^r_{e_1}$ is also a transformation matrix between frames $R_{e_2}$ and $R_r$. Then, multiply the inverse of $H^r_{e_2}$ left by (12) to obtain (13).

$$\tilde{P}^r_i = H^{e_2}_{e_1} P^\beta_i \tag{13}$$

where $H^{e_2}_{e_1} = (H^r_{e_2})^{-1} H^r_{e_1}$. The points $\tilde{P}^r_i$ must be on the plane $\alpha$ according to (13). The preceding analysis demonstrates that the workpiece can be moved above the base plate when the transformation matrixes are obtained, i.e., the workpiece can be moved into the assembly node. The above analysis demonstrates the efficacy of the visual servoing control method proposed in this research based on perspective transformation.

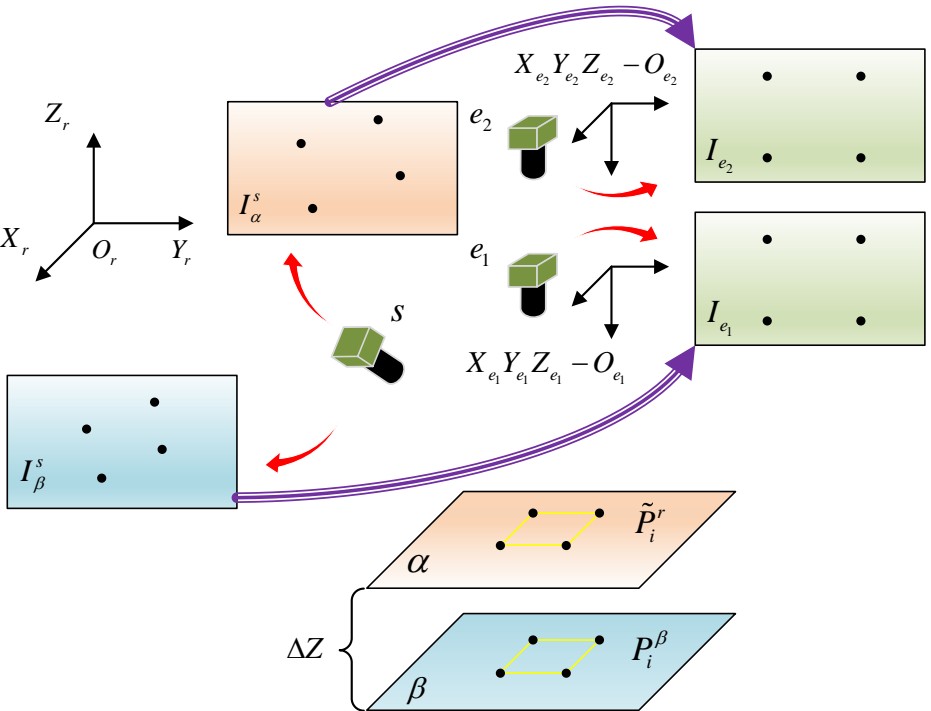

**Figure 3.** The relationship between camera pose and viewing field.

### 3.3. Calculation of the Transformation Matrix

Although the transformation matrix is defined in (12), it comprises not only the intrinsic and external matrixes of the camera but also multiple scaling factors, making it difficult to calculate the matrix directly. Suppose the geometric relationship between several spacial points is known in advance. The transformation matrix can be generated using the direct linear transformation (DLT) method in conjunction with the pre-selection of calibration points. The method is detailed below.

(1) Creating a square with a side length of $d$ and retrieving all of its corners points $P_i$, where $P^u_i$ represents the four upper corner points, and $P^d_i$ represents the four lower corner points. The corresponding image points $p^u_i$ and $p^d_i$ can be extracted using the image processing approaches.

(2) A virtual image plane is created, and four image points $\tilde{p}_i$ forming a square are selected.

(3) The transformational matrix $H^r_{e_1}$ and $H^r_{e_2}$ can be obtained by substituting $p^u_i$, $p^d_i$ and $\tilde{p}_i$ DLT method, respectively.

*3.4. Docking Trajectory Planning*

There are two different methods for docking trajectory planning. The first way is known as the conventional method. When the workpiece reaches the assembly node, each transformation matrix is replaced with a unit matrix with the same dimension. A conventional IBVS controller drives the robotic arm to place the workpiece in the desired position. The other method can be called the attitude extraction method. When the workpiece reaches the assembly node, the attitude of the end-effector can be obtained immediately. The workpiece is moved to the desired position by driving the robotic arm along a linear trajectory within this attitude. Both methods are capable of performing the assembly task in theory. Although future motions are unaffected in the first method when the workpiece does not entirely reach the assembly node, the robotic arm cannot be driven to follow a straight trajectory due to the absence of constraint. The second method ensures that the robot arm's trajectory is always straight, notwithstanding the possibility of assembly failure due to the lack of a correcting mechanism. Taken together, we chose the attitude extraction method overall, and one of the future goals is to improve the proposed method.

## 4. Design of Visual Servoing Controller Based on ADRC

*4.1. Image Features Selection*

Image features are critical for the visual servoing system to perform effectively, and research into practical image features is a significant focus of visual servoing technology. Although a variety of simple image features have been successfully used in the IBVS, these image features are inherently flawed. For instance, decoupling control is difficult when a controller employs point features, whereas linear features are insensitive to displacement. If numerous simple features can be combined to generate a composite image feature that retains all of the advantages of the simple image features, the system's dynamic performance will be excellent [9]. Thus, this research chooses a composite image feature $f(x, y, A, l_1, l_2, l_3, l_4)$ consisting of image moment, area, and lines. The first two components $(x, y)$ are the center of mass used to describe object location. The center of mass is expressed by the first moment, which is the result of all points' joint action, and the role of each image point at the moment is limited. When some image points are affected by noise, it does not bring a significant fluctuation to the moment. The commonly used equation for the center of mass is shown in (14).

$$\bar{x} = \frac{\sum_{i=0}^{m} f(x_i, y_i) x_i}{\sum_{i=0}^{m} f(x_i, y_i)}, \bar{y} = \frac{\sum_{j=0}^{m} f(x_i, y_i) x_i}{\sum_{j=0}^{m} f(x_j, y_j)} \tag{14}$$

The third component $A$ is an area value. The area size is used to achieve position control when combined with image moments since it is susceptible to object depth. It is because the area reflects the "large near and small far" rule of objects. The last four components $(l_1, l_2, l_3, l_4)$ are line parameters used to describe object rotation. The pose is the most challenging state to control in visual servoing due to the solid non-linear relationship between image features and spatial poses. This means we need to explore visual features that can make the robotic trajectory smoother and more robust. Many image features have been successfully used to describe the pose, and in this research, the linear feature is used to describe the pose as the most suitable. The reason for this is that the linear feature has the highest sensitivity to the pose [9], which is beneficial to improve the performance of the closed-loop system, and lines are less challenging to obtain. The linear equation can be shown in (15).

$$x \cos \theta + y \sin \theta = \rho \tag{15}$$

where $\theta$ is an angle between a line and $x$ axis, and $\theta$ is the minimum distance from the origin to the line. The Jacobi matrix based on the linear feature can be expressed as [9]:

$$\dot{\theta} = \begin{bmatrix} L_{\theta vx} & L_{\theta vy} & \frac{1}{2z_{c2}} - \frac{1}{2z_{c2}} & -\rho_1 \cos\theta & -\rho_1 \cos\theta & 1 \end{bmatrix} [v_{ca}\omega_{ca}] \tag{16}$$

### 4.2. Controller Design

The active disturbance rejection controller (ADRC) is a non-linear uncertain system control method based on the traditional proportional-integral-differential (PID) control method and modern control theory [32]. An ADRC should be consists of a tracking differentiator(TD), a control law of non-linear state error feedback (NLSEF), and an extended state observer(ESO). The earliest application of ADRC to visual servoing technology was by Jianbo Su [33]. The main idea of ADRC in an IBVS system is to first create a linear system model with a constant Jacobi matrix and then use the ESO and NLSEF to estimate and dynamically compensate for non-linear model and positional errors [34]. Figure 4 shows the schematic of an ADRC-based visual servoing system.

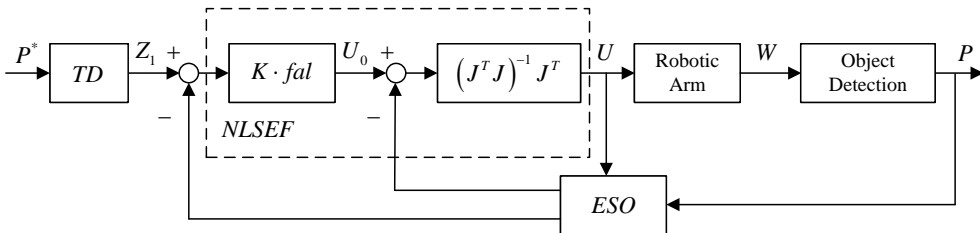

**Figure 4.** Control system block diagram

Figure 5 illustrates the structure of the composite visual feature-based ADRC used in this research. The controller receives the composite image features $p_i^*$ as input, and the output $u_i$ are six-velocity signals for the robotic arm. Next, the ADRC controller is described in the following three aspects. A second-order TD chosen in this research is shown in (17).

$$\begin{cases} \varepsilon_1 = z_1(k) - \tilde{p}_1(k) \\ z_1(k+1) = z_1(k) - h \cdot r \cdot fal(\varepsilon_1, \alpha_1, \delta_1) \end{cases} \tag{17}$$

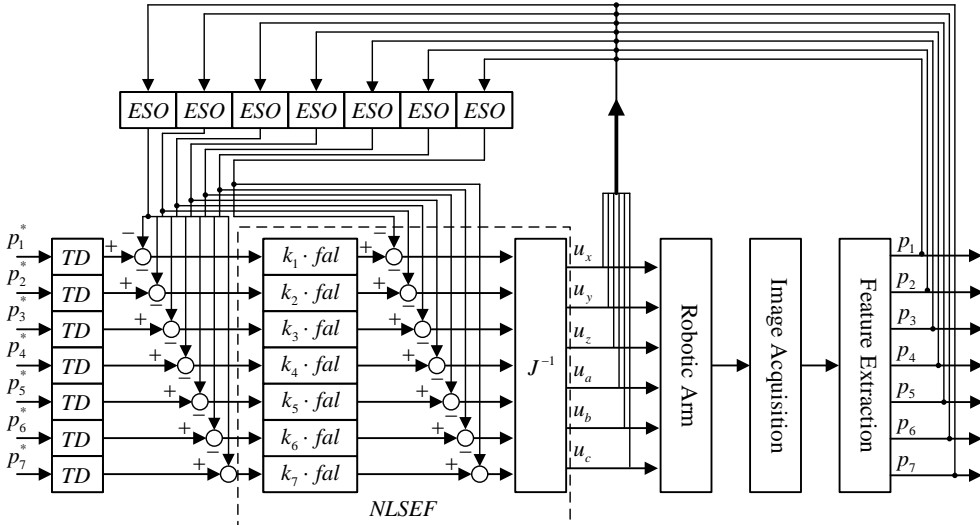

**Figure 5.** Control system block diagram.

There are two parameters to be adjusted in (17), which are the speed factor $\alpha_1$ and the filtering factor $\delta_1$. $\alpha_1$ determines the tracking speed. The larger the value, the faster the

tracking speed of TD; $\delta_1$ determines the tracking accuracy. The larger the value, the higher the tracking accuracy of TD. Since TD has better independence, parameter adjustment can be performed offline according to the object's needs. The ESO is designed as shown in (18).

$$\begin{cases} \varepsilon_2 = z_2(k) - p_2(k) \\ z_2(k+1) = z_2(k) - h(z_3(k) - b_1 \cdot fal(\varepsilon_2, \alpha_2, \delta_2) + J_i \cdot u(k+1)) \\ z_3(k+1) = z_3(k) - h \cdot b_2 \cdot fal(\varepsilon_2, \alpha_3, \delta_3) \end{cases} \tag{18}$$

where $\varepsilon_2$ are tracking errors, $z_2$ is tracking signal, $z_3$ is the observation of disturbance; $b_1$ and $b_2$ are gains of ESO, $\alpha_2$ and $\alpha_3$ are nonlinear factors which values in the range $(0, 1]$.The larger the values of $\alpha_2$ and $\alpha_3$, the greater the nonlinearity of $fal$. $\delta_2$ and $\delta_3$ are the widths of the linear interval of the function $fal$.

The NLSEF is designed as shown in (19), and it employs non-linear feedback of errors instead of linear feedback in classical PID, which is a highly efficient control strategy. The non-linear calculation of the visual characteristic error signal using NLSEF can improve the control accuracy and robustness of the system.

$$\begin{cases} \varepsilon_3 = z_1(k) - z_2(k) \\ u = h \cdot k \cdot fal(\varepsilon_3, \alpha_3, \delta_3) \\ u(k) = J^{-1} \begin{pmatrix} u_1 - z_3^1(k) \\ \vdots \\ u_6 - z_3^6(k) \end{pmatrix} \end{cases} \tag{19}$$

where $\varepsilon_3$ are tracking errors, $\alpha_3$, $\varepsilon_3$, and $\delta_3$ are three factors of $fal$. $J$ is the image jacobi matrix and $u_i$ are desired values.

## 5. Simulation

### 5.1. Simulation Parameters

To demonstrate the efficiency of the proposed method, a visual servoing simulation system was built using a six-degree-of-freedom (DOF) robotic arm model coupled with an eye-to-hand (ETH) camera configuration. The camera was used to project numerous spatial points into the image plane, and the parameters listed in Table 1 do not apply to the visual servoing controller. For convenience, the lens distortion of the camera was neglected.

**Table 1.** Simulation Parameters of the Camera.

| Parameter | Value |
|---|---|
| Focal length | 0.008 |
| Length | 1024 |
| Width | 1024 |
| Coordinates of the projection center | (512,512) |
| Scaling factors | (0.00001,0.00001) |

A transformation matrix of the robotic end-effector frame to the camera frame is

$$T_e^c = \begin{pmatrix} -1 & 0 & 0 & 0.01 \\ 0 & -1 & 0 & 0.01 \\ 0 & 0 & 1 & 0 \\ 0 & 0 & 0 & 1 \end{pmatrix} \tag{20}$$

The transformation matrixes $H_1$ and $H_2$ were obtained by the DLT method.

$$H_1 = \begin{pmatrix} 1.25 & -1.25 & 350 \\ -1.25 & -1.25 & 1570 \\ 0 & 0 & 1 \end{pmatrix} \tag{21}$$

$$H_2 = \begin{pmatrix} 1.0937 & -1.0937 & 350 \\ -1.0937 & -1.0937 & 1570 \\ 0 & 0 & 1 \end{pmatrix} \tag{22}$$

The simulation parameters in the ADRC controller are listed in Table 2.

**Table 2.** Parameters of The ADRC.

| Part | Parameters | Value |
|---|---|---|
| TD | $h$ | 0.1 |
| | $\alpha_1$ | 0.02 |
| | $\delta_1$ | 0.12 |
| ESO | $\alpha_2$ | 0.5 |
| | $\delta_2$ | 0.5 |
| | $\alpha_3$ | 0.01 |
| | $\delta_3$ | 60 |
| | $\gamma$ | 1200 |
| | $b_1$ | 15 |
| | $b_2$ | 0.7 |
| NLSEF | $\alpha_4$ | 0.5 |
| | $\delta_4$ | 10.5 |

In the simulation, the step of image processing was skipped. Four coplanar spatial coordinates $P_i$ with fixed relationships were used to represent a workpiece, with the starting and desired positions in the world and image frames given in Table 3. In the world frame, the coordinates of $P_i$ were unknown. However, the servoing controller knows the projection coordinates of $P_i$ in the image plane.

**Table 3.** Starting and Desired Positions of Feature Points In The Simulation.

| Position | Num | Spatial Point | Image Point |
|---|---|---|---|
| Starting Position | 1 | $(-0.48, -0.81, 1.83)$ | $(301.82, 158.65)$ |
| | 2 | $(-0.81, -0.08, 1.89)$ | $(169.46, 476.81)$ |
| | 3 | $(-0.12, 0.21, 2.17)$ | $(468.07, 588.85)$ |
| | 4 | $(0.21, -0.52, 2.11)$ | $(592.84, 315.47)$ |
| Desired Position | 1 | $(0.2, 0.2, 5)$ | $(544, 544)$ |
| | 2 | $(0.2, 1, 5)$ | $(544, 672)$ |
| | 3 | $(1, 1, 5)$ | $(672, 672)$ |
| | 4 | $(1, 0.2, 5)$ | $(689.24, 544)$ |

The steady-state error, which can be defined as (23), was used to evaluate the controller's performance for point-to-point control. $\tilde{p}_i$ and $p_i$ are the projections of the desired and current positions in the image plane, respectively. When the positioning error between the current and desired positions is less than 0.05 pixels in the image, we consider the workpiece to have reached the assembly point.

$$E = \sum_{i=1}^{4} |p_i - \tilde{p}_i| \tag{23}$$

### 5.2. Simulation and Discussion

In the first simulation, the efficiency of the proposed visual servoing control method based on perspective transformation was evaluated, and the results are shown in Figure 6. The spatial trajectory of the robotic arm is smooth, with no cases of camera retreat. The position error drops dramatically within the first 100 iterations, reaching a threshold of 0.05 pixels by the 158th iteration. As illustrated in the simulation results, the control method

proposed in this research accomplishes the desired goal of delivering the workpiece to the assembly node in an attitude.

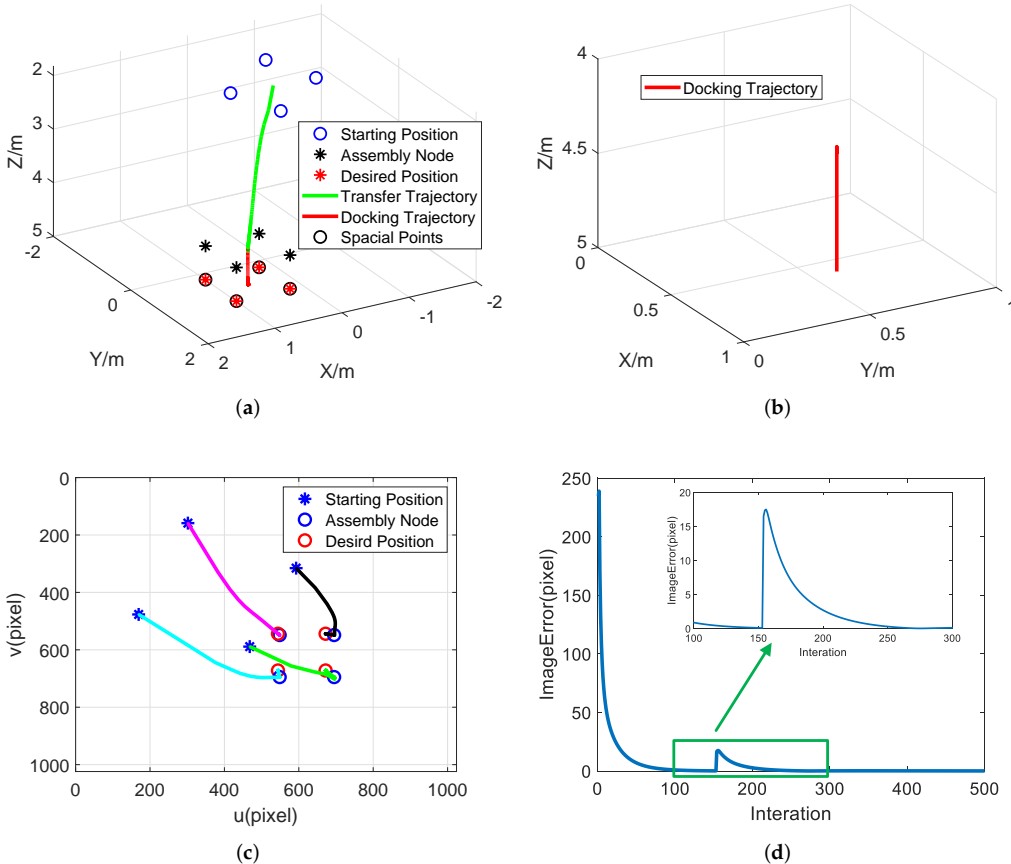

**Figure 6.** The First Simulation Result by The Proposed Method: (**a**) Complete Spatial trajectory of the robotic arm. (**b**) Docking trajectory. (**c**) Image trajectory. (**d**) Position error in the image.

In the second simulation, on the other hand, the transformation matrixes $H_1$ and $H_2$ were replaced by unit matrixes of the exact dimensions while all other simulation settings remained unchanged. The results shown in Figure 7 demonstrate that the trajectory of the robotic arm is generally smooth, with few large fluctuations. However, $P_i$ reaches the target position immediately and does not pass via the assembly node, demonstrating that the conventional control method cannot resolve the spatial constraint problem.

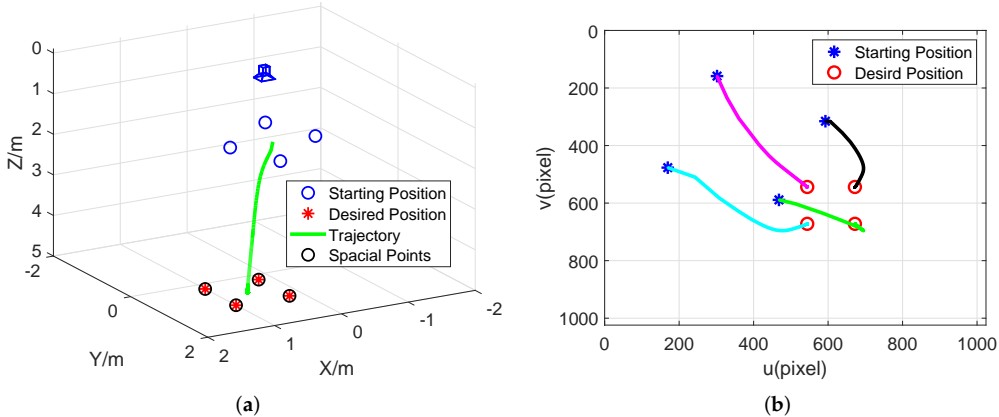

**Figure 7.** Simulation results by the classical controller: (**a**) Complete Spatial trajectory of the robotic arm. (**b**) Image trajectory.

The third simulation was performed to ensure that the method used in Section 3.4 is valid. When the workpiece reaches the assembly node, $H_1$ and $H_2$ are replaced with unit matrixes of the same size, and the docking trajectory is planned using the conventional method based on ADRC, as illustrated in Figure 8. Despite the small distance between the assembly node and the destination position, the conventional controller has difficulty driving the robotic arm in a straight trajectory, as demonstrated in Figures 6 and 8. The assembly task cannot be completed in this situation, even though the workpiece has been transported into the assembly node.

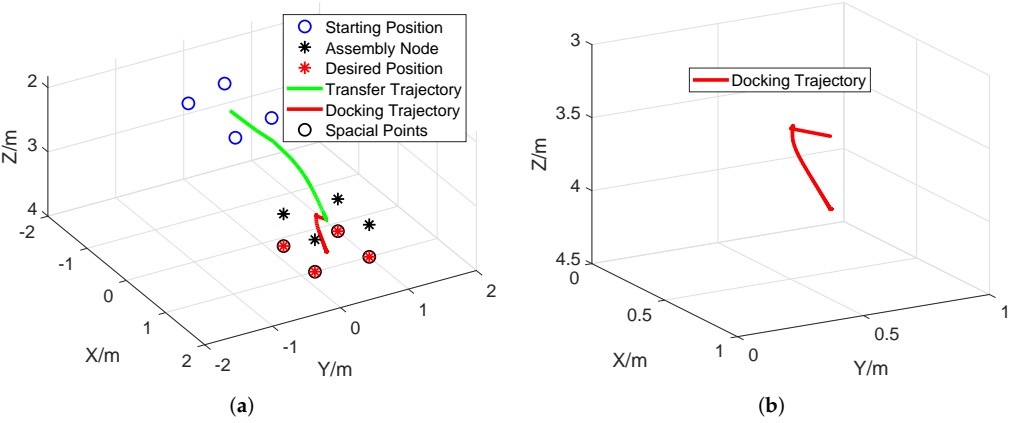

**Figure 8.** Simulation results by the classical controller: (**a**) Complete Spatial trajectory of the robotic arm. (**b**) Docking trajectory.

The fourth simulation, in which the point features were used to create the image Jacobi matrix and other parameters were intact, was performed to demonstrate the superiority of the composite visual features. The result is shown in Figure 9. Although the point feature-based ADRC controller can drive the arm to the desired position, the trajectory is not smooth since the workpiece is first moved in the direction of increasing position error and then returned to normal. This behavior is caused by the absence of uncoupling capacity in the point-based controller, which increases the probability of spatial points becoming displaced from the FOV of the camera. Furthermore, the steady-state error of the first simulation converges faster than the fourth, demonstrating that the composite image feature is more sensitive to robotic arm motion.

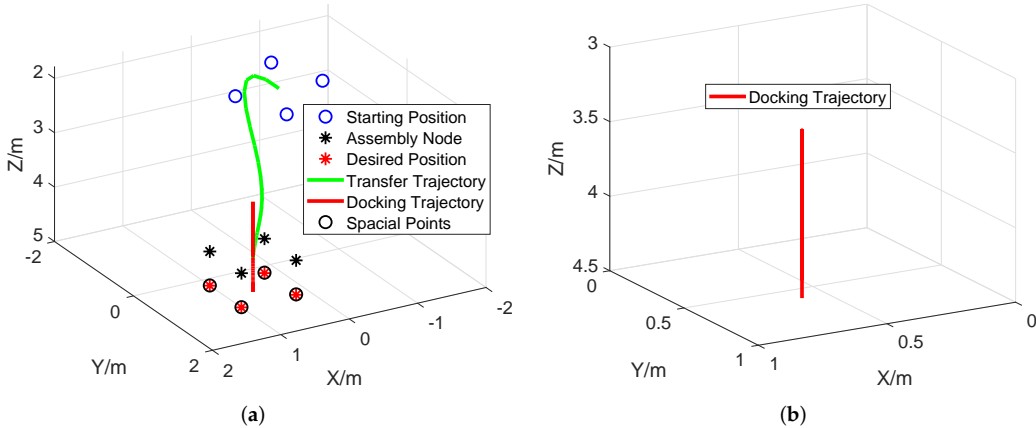

**Figure 9.** *Cont.*

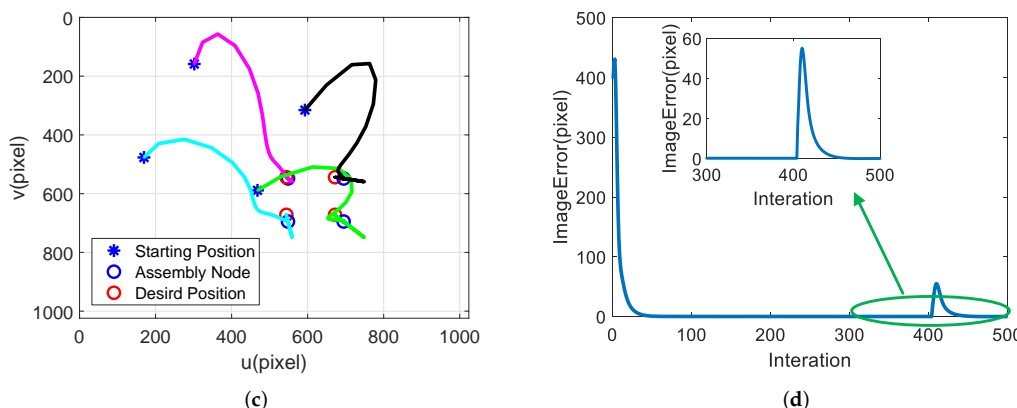

**Figure 9.** Results of the fourth simulation: (**a**) Complete Spatial trajectory of the robotic arm. (**b**) Docking trajectory. (**c**) Image trajectory. (**d**) Position error in the image.

Finally, a simulation based on the PID controller was conducted to demonstrate the advantages of the ADRC controller in IBVS, with the result shown in Figure 10. The position and attitude of the workpiece are continually modified as it approaches the assembly node. The results demonstrate how difficult it is to achieve satisfactory performance for a strongly coupled visual servoing system using a conventional PID controller.

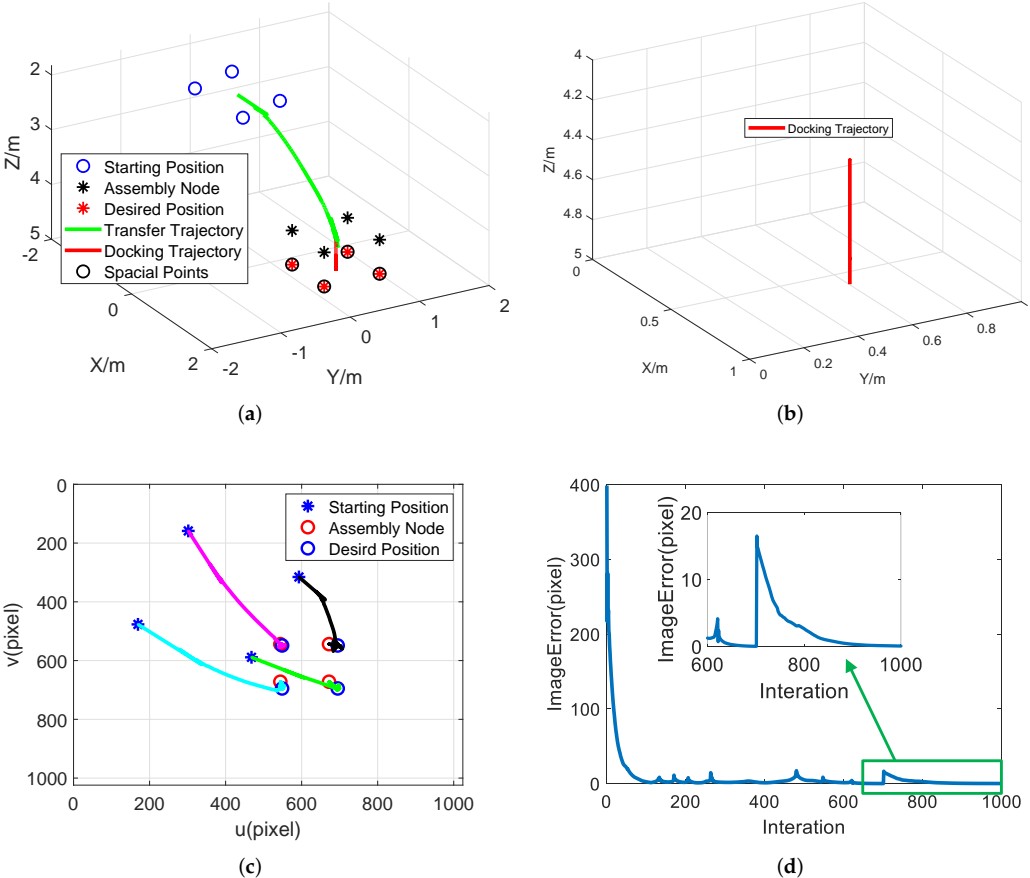

**Figure 10.** Results of the last simulation: (**a**) Complete Spatial trajectory of the robotic arm. (**b**) Docking trajectory. (**c**) Image trajectory. (**d**) Position error in the image.

## 6. Experiment and Discussion

As demonstrated in Figure 11, a visual servoing system is established. The experimental setup consists of a camera, a computer, and a robot (Dobot Magician) with four DOF. The software was coded by MFC with OpenCV 3.2 and consisted of three parts: (1) a camera and a robot control system, which includes initialization, start and stop functions, and parameter setting; (2) a real-time display system consisting of an image display and information display; and (3) an information storage system, which was designed to save important data throughout the program operation. The camera is MER-200-14GC with a 12-mm lens and a $1628 \times 1236$ image resolution. The robotic arm has a repeated positioning accuracy of 0.2 mm and a minimum movement distance of 0.05 mm. As illustrated in Figure 12, the workpiece is a white card with four colored circles, and the base plate is a slot the same size as the white card. The inside slot is also marked with colored circular marks, ensuring that the relationship between the white card and the slot meets the space constraint.

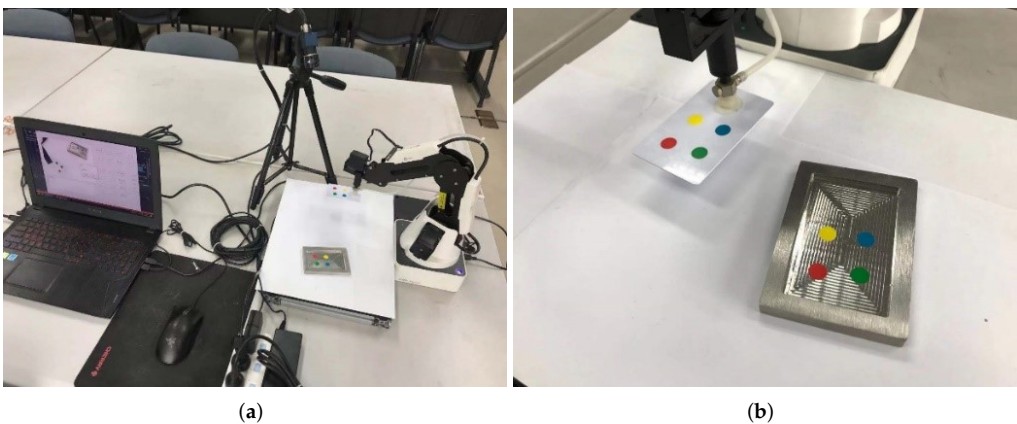

(a)  (b)

**Figure 11.** Experimental setup: (**a**) Complete experimental platform. (**b**) The white card and the slot.

The homography matrixes mentioned in Section 3.1 were produced using a customized calibration block with two parallel plates, as shown in Figure 12a. Each plate has four incomplete circular regions, the centers of which create a cuboid with known dimensions in space. The most significant advantage of the calibration block is that it allows the camera to observe all circular regions simultaneously without obstruction. Figure 12b depicts an aluminum calibration block with a production error of less than 0.1 mm, and Figure 12c depicts the exact dimensions of the calibration block. All circular regions were marked with different colored stickers, and the purple sticker was used to differentiate the plate.

$$H_1 = \begin{bmatrix} 2.76 & 0.67 & -2163.40 \\ 0.66 & -4.08 & 719.50 \\ -1.07 \times 10^{-5} & 0.00037 & 1 \end{bmatrix} \tag{24}$$

$$H_2 = \begin{bmatrix} 3.15 & 0.69 & -2554.26 \\ 0.67 & -4.28 & 1730.30 \\ -2.22 \times 10^{-5} & 0.00032 & 1 \end{bmatrix} \tag{25}$$

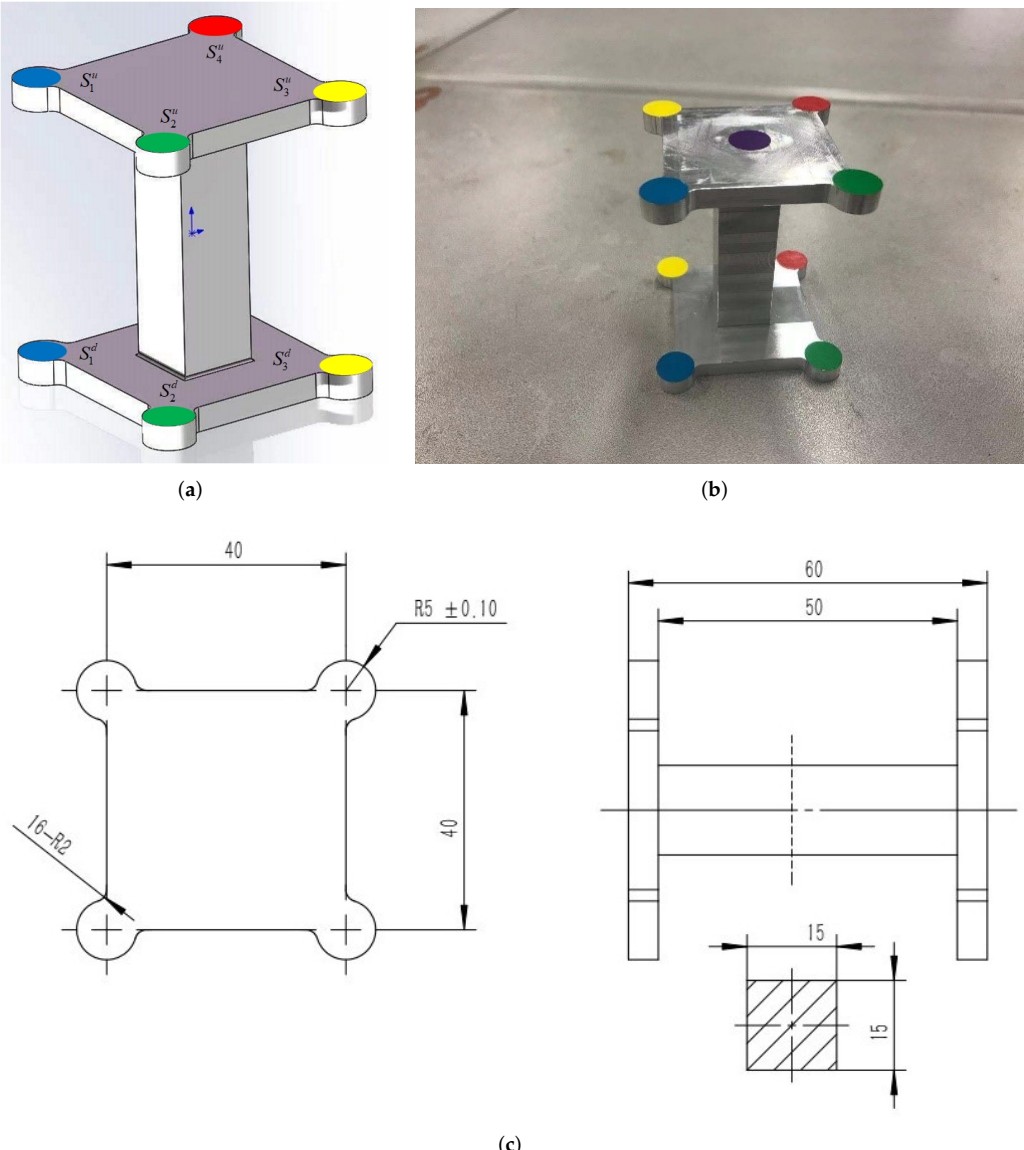

**Figure 12.** Calibration block: (**a**) Design drawing. (**b**) The aluminum calibration block. (**c**) Exact dimensions of the calibration block.

The first test was carried out; the results are shown in Figure 13. The starting position of the robotic arm is depicted in Figure 13a, and it then begins to move toward the assembly node, driven by the controller. The robotic arm has reached the assembly node in Figure 13b, and the white card is just above the slot. The card's position continues to descend the Z-axis until it is inserted into the slot. However, the card cannot reach the bottom of the slot since the surface of the lower plate does not coincide with the genuine working plane. As a result, a pressure sensor was fitted at the robotic arm's end. When the sign of the measured value changes, it is indicated that the workpiece has arrived at the predetermined point.

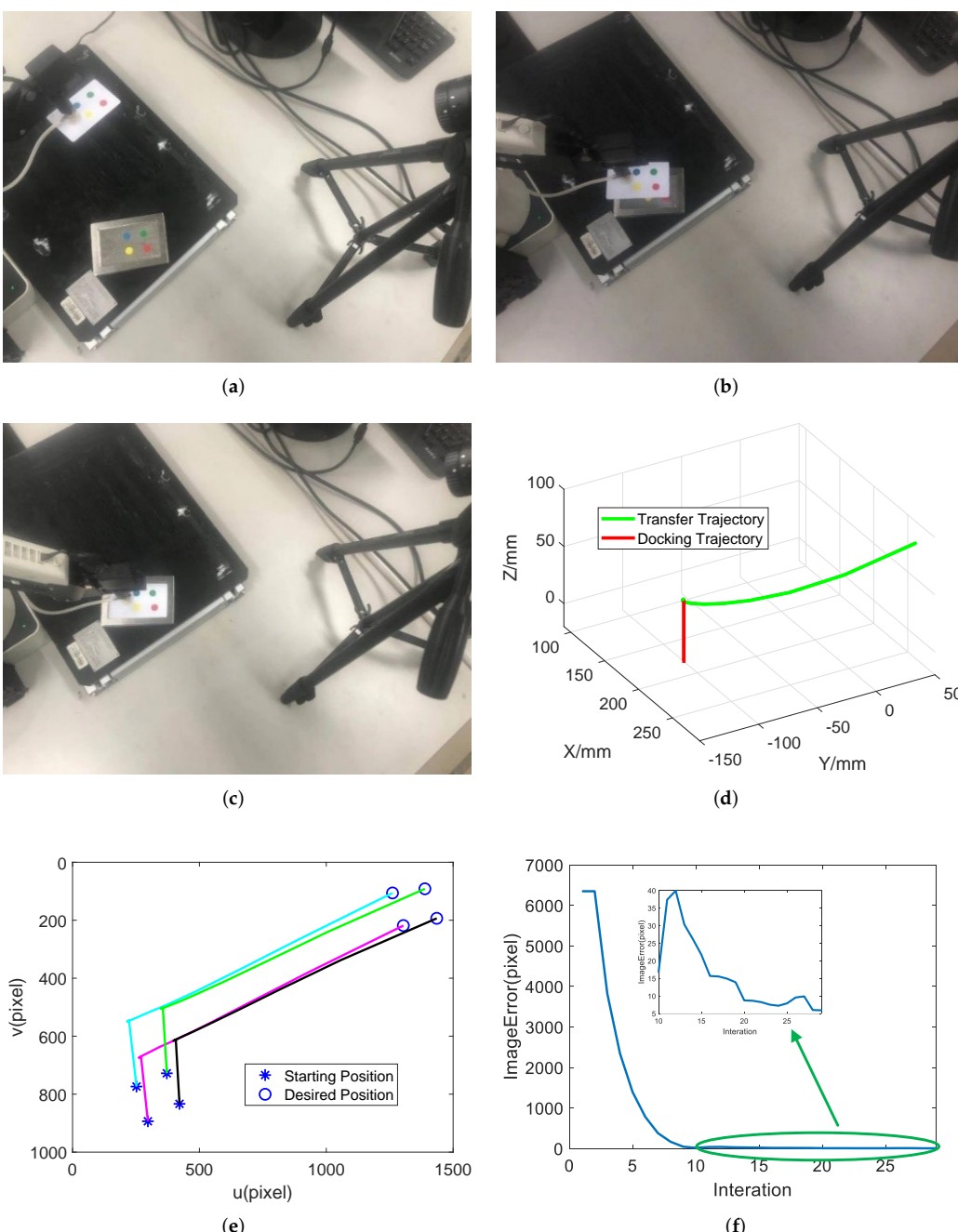

**Figure 13.** Experimental result of visual servoing control method based on perspective transformation: (**a**) Starting position of the robotic arm. (**b**) The robotic arm reached the assembly node. (**c**) The assembly task is accomplished. (**d**) Complete Spatial trajectory of the robotic arm. (**e**) Image trajectory. (**f**) Position error in the image.

Another test was performed in which the transformation matrixes $H_1$ and $H_2$ were replaced with unit matrixes while the other experimental settings remained the same. Figure 14 shows the outcomes of this test. As seen in Figure 14b, the card has collided with the slot and cannot be inserted into the slot because the gap between the card and the slot is smaller than 1mm. Due to the low movement speed, the robotic arm did not stop moving after colliding with the card, resulting in significant deformation. However, if a collision occurs during an assembly, the robotic arm will be damaged, which is not permitted in an assembly operation.

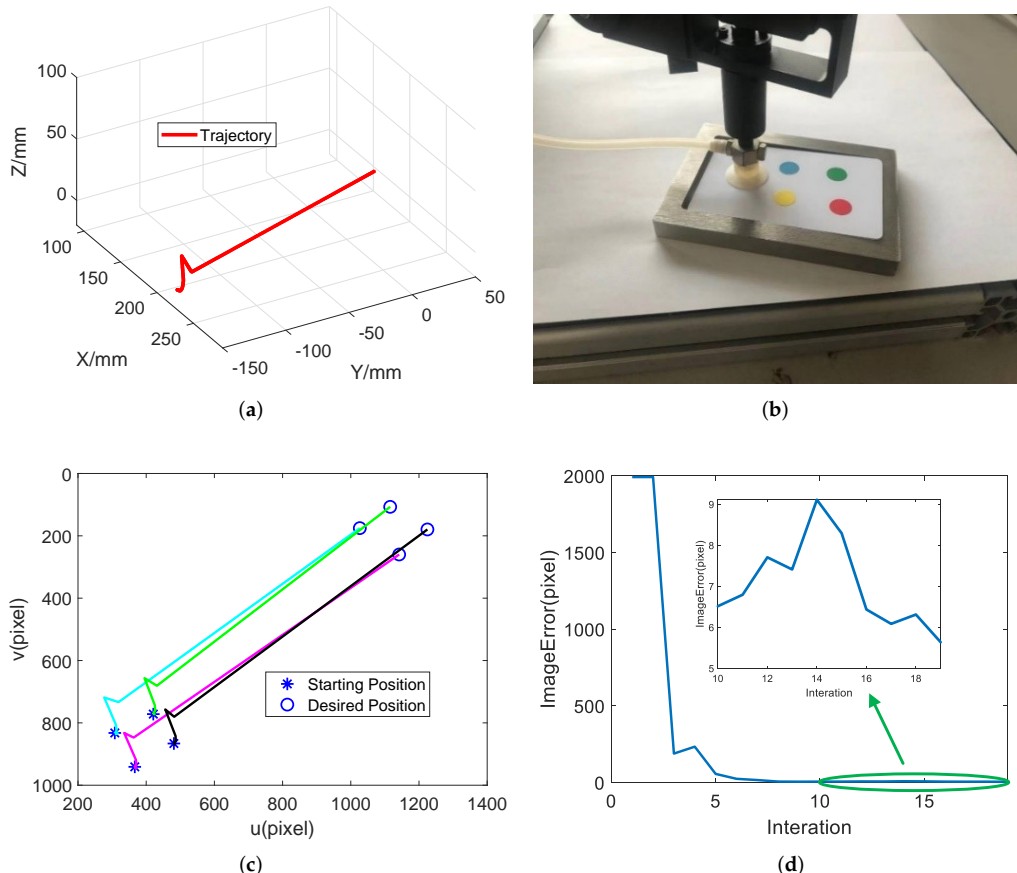

**Figure 14.** Experimental result of conventional method: (**a**) Starting position of the robotic arm. (**b**) The collision between the card and the slot. (**c**) Image trajectory. (**d**) Position error in the image.

Experiments are repeated 50 times to ensure that the presented method works and the results are shown in Table 4. As a comparison, 50 experiments were conducted using the conventional IBVS controller. The method proposed in this research has a 100% success rate and a position accuracy of less than 1 mm. On the other hand, the collision problem described in Figure 14, on the other hand, occurs in all assembly experiments using the conventional IBVS controller. This fully demonstrates that the method proposed in this research effectively solves the problem of spatial constraint.

**Table 4.** Results of 50 experiments.

| Terms | The Proposed Method | Conventional IBVS Method |
|---|---|---|
| Total Times | 50 | 50 |
| Successful Times | 50 | 0 |
| Average Error | <1 mm | 1.69 mm |
| Time Consumption | 17.45 s | >40 s |

## 7. Conclusions

This study presents a visual servo control method based on perspective transformation to transport a workpiece to an unmarked spatial position using a robotic arm under uncalibrated conditions. A customized calibration block with two parallel plates was created, and all circular areas' centers were used to generate two transformation matrixes. Following that, a virtual image plane was created using the two matrixes. Then projections of the target position and workpiece were obtained in the virtual image plane. Finally, a composite visual feature-based ADRC controller was built to increase the robotic arm system's performance. The workpiece was successfully moved to the desired position by

tracking a given image feature in the virtual plane. The experiment findings indicate that the method proposed in this work achieved a success rate of 100% and a position precision of less than 1 mm, which meets the assembly task requirements.

**Funding:** This research received no external funding.

**Data Availability Statement:** The data generated and/or analyzed during the current study are not publicly available for legal/ethical reasons but are available from the corresponding author on reasonable request.

**Conflicts of Interest:** The authors declare no conflict of interest.

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
