# Peer review of "Research on a Visual Servoing Control Method Based on Perspective Transformation under Spatial Constraint"

_machines, doi:10.3390/machines10111090_

Round 1
Reviewer 1 Report
This paper proposes a method for controlling the end-effector of a robotic arm to a spatial point without a coordinate. The method is a two-stage visual servoing control approach based on perspective information. The method is tested in simulations on a 6DOF robot and a 4DOF of real robot.
Overall, this paper is well written and almost to a level such that it can be accepted. I have a couple of minor remarks that I hope will be addressed by the author:
1) I am not an expert in the field, but it seems to me that the list of references that deal and propose novel methods for this problem is limited. Please verify if all relevant references are cited.
2) The introduction has a number 0. I would use 1 as this is more standard.
3) “…of the robotic arm to more closely to a straight line to avoid…” Please check that phrase.
4) “…, then a controller drive the robotic…” should be drives.
5) In figure 2, the authors make use of colors. For clarity, it could be better to use markers since if printed in gray colors, the color differentiation does not work very well.
61) Under (2), how is the matrix Mc defined.
22) In section 2.3: “… also multiple scaling factors, making it difficult to calculate…”. How are these scaling factors defined?
33) “Suppose the geometric relationship between several special points is known in advance.” Is this a realistic assumption?
34) Under Figure 3, (3) there is twice the same matrix defined.
15) In Section 3.2: “An ADRC is consist of a tracking…” should be consists of..
16) Could the authors comment a bit more on the TD and NLSEF controllers? How are the controller’s parameters defined?
17) In Fig. 4, are u_i the velocity for each joint? More explanation about Figure 4 is desired. Many variables in this figure are not well explained.
18) The simulation study is done on a 6DOF robot, while the real setup contains 4DOF. Is it more interesting to also do a simulation with a 4DOF robot?
19) In Section 4, the distortion on the camera is neglected. How realistic is this? How could it be included and what could be the effect on the results? Please comment more on the assumptions.
110) Above (14), the DLT method is used to compute the transformation matrices. Be more specific about how these matrices are computed. Refer to equation or somewhere such that the reader can more easily verify the work.
111) In Table 2, the parameters of the controllers are given, but I have no idea what they mean. The controller’s structure is not given. Be more specific please.
112) “In the simulation, the step of image processing was skipped,..” How realistic is this assumption? Please comment on these assumptions.
113) “However, the servoing controller is aware of the projection coordinates of P_i in the image plane.” How realistic is this assumption? Please comment on these assumptions.
114) In Section .2, the authors talk about iterations. I don’t get where these iteration come from. Please explain better what equation needs to be iterated.
Author Response
We appreciate the time and effort you dedicated to providing feedback on our manuscript and are grateful for the insightful comments and valuable improvements to our paper. We have incorporated all the suggestions made by the reviewers. Those changes are highlighted in the manuscript.
1) This article was completed in June 2021, so I reviewed new articles from the last year and added several relevant literatures to the references.
2) Thanks for your suggestion. The number has changed to 1.
3) I have rewritten the sentence to make its meaning clearer
4) Thanks for your suggestion. I have changed.
5) Thanks for your suggestion. I have added more arrows and logos to the image so that the reader can understand the meaning of figure2 very clearly.
61) I added the definition of Mc in the article
22) These scale factors are , , , and , and they are determined by the position of camera.
33) Yes, when we get a workpiece, the size of the workpiece is known.
34)Thanks, I have corrected it
15)Thanks, I have corrected it
16)ADRC is a controller proposed by Han Jingqing, a Chinese scholar. In this paper, we just used ADRC and did not optimize it, so we did not introduce it carefully. I have added two references and a description of the ADRC for the readers who need them.
17)I have added a partial description to the article
18) This is a regrettable thing. We wanted to conduct experiments on a 6DOF robot arm, but our lab could not afford a 6DOF robot, so we had to finish the experiments on a 4DOF robot.
19) In visual servoing and visual measurement, distortions must first be corrected to weaken the measurement error. A commonly used correction method is the general plane-based calibration proposed by Z. Zhang. The effect of lens distortions on visual servoing is mentioned in the paper.
[1] Z. Zhang, “Flexible camera calibration by viewing a plane from unknown orientations,” in The Proceedings of the Seventh IEEE International Conference on Computer Vision (IEEE, 1999), Vol. 1, pp. 666–673.
[2] Gong Zeyu, Tao Bo, Yang Hua, Yin Zhouping, Ding Han. An Uncalibrated Visual Servo Method Based on Projective Homography. IEEE Transactions on Automation Science and Engineering, 2018, 15(2): 806-817
110) The DLT method is a simple and effective way of completing hand-eye calibrations. It is described in detail in my previous paper, to which I have added a reference.
112) This paper aims to deal with spatial constraints, and therefore, image processing-related content is omitted. For target detection and semantic segmentation, many methods are already available, encompassing both traditional and deep learning-based approaches.
113) This means that with image processing basics, we can know the coordinates of the image point of a spatial point P. It is an effortless step to implement
114) Iteration is the number of loops in the simulation.
Reviewer 2 Report
This paper presents a two-stage visual servoing control approach based on perspective transformation for controlling a robotic arm.
The paper, in general, is well-organized and well-written. The simulation and real-world experiments have been presented to validate the methodology.
However, some part of the methodology needs more elaboration.
- One of the essential things missing is the actual visual servoing control law to compute the joint velocity used for the simulation and experiment. The author only mentions that the controller receives the composite image features as input, and the output is a velocity signal for the robotic arm. More elaboration and formulation are required, starting from the image features to the four DOF joint velocity derivations. This will definitely enhance the contribution of this work.
- The convergence and stability of the proposed IBVS system have to be discussed.
- The discussion of ADRC in an IBVS system is limited. More explanation or proper references should be added.
- The four DOF of a robot (Dobot Magician) must be clearly mentioned in the paper.
- It is mentioned that line features are employed for orientation. It is better to have more explanation of the line parameters used.
- The composite feature and seven-element vector need to be explained in more detail. What are image moments? How can an area value indicate an object’s depth in the world frame?
- It is reported that the Proposed Method has 100% success, whereas the Conventional IBVS Method has 0% success. More reasoning and discussion are required for this.
- Some Corrections:
- Section 6 => Conclusion instead of conclusion.
- Abstract => active disturbance rejection controller (ADRC) instead of ARDC only.
Author Response
We appreciate the time and effort you dedicated to providing feedback on our manuscript and are grateful for the insightful comments and valuable improvements to our paper. We have incorporated all the suggestions made by the reviewers. Those changes are highlighted in the manuscript.
- Thanks for your suggestion. I have added descriptions of ADRC and composite visual features.
- Your suggestion is correct, but my research does not involve control theory, so I could not figure out the proof process of ADRC. For this reason, I have only added a few references related to ADRC to my research.
- I have added a detailed description of the ADRC principle.
- The four DOF of a robot (Dobot Magician) was clearly mentioned in the paper.
- Thanks for your suggestion. I have added descriptions of ADRC and composite visual features.
- I have added a description of the image moments.
- We illustrate in section 2 that the traditional IBVS method cannot solve the spatial constraint problem, so the success rate of the traditional method under the spatial constraint must be 0. Our proposed method is specifically designed to solve the spatial constraint problem.
- I have finished the corrections.
Reviewer 3 Report
Aiming at the improvement of the flexibility and accuracy of the manipulator, the paper introduces assembly nodes and establishes corresponding spatial constraints, and proposes a two-stage visual servoing control method based on virtual image plane and perspective transformation. Then designs experiments to verify the effectiveness of the proposed method. Overall, this is an impressive work even though there are a few issues to address.
Issues to address:
The existing related methods are introduced in this paper, but most of the methods mentioned and references cited are years ago. Hope to introduce and analyze more state-of-art methods and compare them with the methods proposed in this paper.
Section 3.1 mentions the use of composite image features, including image moments, area sizes and lines, but does not describe the methods used to extract these features, and how to ensure the stability of feature tracking?
The description in the "Controller Design" section is too short. Hope to provide a more detailed description of the controller shown in Figure 4.
In lines 47-48 of the paper, it is mentioned that the MPC method faces the difficulty of trajectory modification in the presence of obstacles. Can the method proposed in this paper effectively solve this problem?
Author Response
We appreciate the time and effort you dedicated to providing feedback on our manuscript and are grateful for the insightful comments and valuable improvements to our paper. We have incorporated all the suggestions made by the reviewers. Those changes are highlighted in the manuscript.
1. This article was completed in June 2021, so I reviewed new articles from the last year and added several relevant literatures to the references.
2 & 3. Thanks for your suggestion. I have added descriptions of ADRC and composite visual features.
4. Sorry, the method we propose is for solving the spatial constraint problem and does not solve the obstacle avoidance problem. This is a pity, and we will further solve the obstacle avoidance problem under spatial constraints in our subsequent work.
Round 2
Reviewer 2 Report
The authors have answered all my concerns, some of them not very satisfactorily but in an acceptable way.
Reviewer 3 Report
My concerns have been addressed. I have no comments now.